# Incidence of Lyme disease in the United Kingdom and association with fatigue: A population-based, historical cohort study

**Florence Brellier**[1]*, **Mar Pujades-Rodriguez**[1], **Emma Powell**[2], **Kathleen Mudie**[3], **Eliana Mattos Lacerda**[3], **Luis Nacul**[3,4,5], **Kevin Wing**[2]

**1** IQVIA, London, United Kingdom, **2** Department of Non-communicable Disease Epidemiology, Faculty of Epidemiology and Population Health, London School of Hygiene and Tropical Medicine, London, United Kingdom, **3** Department of Clinical Research, Faculty of Infectious and Tropical Diseases, London School of Hygiene and Tropical Medicine, London, United Kingdom, **4** British Columbia Womens Hospital and Health Centre, Complex Chronic Diseases Program, Vancouver, Canada, **5** Department of Family Practice, University of British Columbia, Vancouver, Canada

* florence_brellier@hotmail.com

**Data Availability Statement:** The data underlying the results presented in the study are from IQVIA Medical Research Data (IMRD), which

## Abstract

### Background

Estimations of Lyme disease incidence rates in the United Kingdom vary. There is evidence that this disease is associated with fatigue in its early stage but reports are contradictory as far as long-term fatigue is concerned.

### Methods and findings

A population-based historical cohort study was conducted on patients treated in general practices contributing to IQVIA Medical Research Data: 2,130 patients with a first diagnosis of Lyme disease between 2000 and 2018 and 8,510 randomly-sampled patients matched by age, sex, and general practice, followed-up for a median time of 3 years and 8 months. Main outcome measure was time to consultation for (1) any fatigue-related symptoms or diagnosis; or (2) myalgic encephalomyelitis/chronic fatigue syndrome (ME/CFS). Adjusted hazard ratios (HRs) were estimated from Cox models. Average incidence rate for Lyme disease across the UK was 5.18 per 100,000 person-years, increasing from 2.55 in 2000 to 9.33 in 2018. In total, 929 events of any types of fatigue were observed, leading to an incidence rate of 307.90 per 10,000 person-years in the Lyme cohort (282 events) and 165.60 in the comparator cohort (647 events). Effect of Lyme disease on any subsequent fatigue varied by index season: adjusted HRs were the highest in autumn and winter with 3.14 (95%CI: 1.92–5.13) and 2.23 (1.21–4.11), respectively. For ME/CFS, 17 events were observed in total. Incidence rates were 11.76 per 10,000 person-years in Lyme patients (12 events) and 1.20 in comparators (5 events), corresponding to an adjusted HR of 16.95 (5.17–55.60). Effects were attenuated 6 months after diagnosis but still clearly visible.

incorporates data from The Health Improvement Network (THIN), a Cegedim database. Authors do not have the permission to share the data. Researchers have the possibility to access IMRD similarly to the authors, subject to a sublicense and an approved protocol. They may contact IMRDEnquiries@iqvia.com for this purpose.

**Funding:** The author(s) received no specific funding for this work.

**Competing interests:** The authors have read the journal's policy and have the following competing interests: Mar Pujades-Rodriguez and Florence Brellier are now employees at Union Chimique Belge (UCB) Biopharma and at Bristol Myers Squibb, respectively. They were both IQVIA employees while working on this study and there are no patents, products in development or marketed products associated with this research to declare. This does not alter the authors' adherence to PLOS ONE policies on sharing data and materials.

## Conclusions

UK primary care records provided strong evidence that Lyme disease was associated with subsequent fatigue and ME/CFS. Albeit weaker on the long-term, these effects persisted beyond 6 months, suggesting patients and healthcare providers should remain alert to fatigue symptoms months to years following Lyme disease diagnosis.

## Introduction

Lyme disease is a tick-borne infection caused by spirochetes of the *Borrelia burgdorferi* sensu lato complex. Infected individuals typically develop an expanding rash starting from the bite location (erythema migrans), as well as flu-like symptoms. Treatment consists of a 2- to 4-week course of antibiotics such as doxycycline, amoxicillin, and cefuroxime. Without antibiotic therapy the disease can spread to the muscles, the joints, and the central and peripheral nervous systems. The mean incidence rate of Lyme disease has been estimated as 56.3 per 100,000 person-years (py) in Western Europe [1], with large variations between countries likely to be due to differences in tick density and burden of tick disease across geographies (from 0.001 in Italy to 464 in Sweden). In England and Wales, 1,579 laboratory-confirmed cases were reported in 2017 [2], corresponding to an incidence of 2.7 per 100,000 py.

There is evidence that Lyme disease is associated with fatigue in its early stage, especially in untreated patients [3]. However, reports are contradictory as far as long-term fatigue is concerned [4–7]. In addition, it is unclear whether Lyme disease could be associated with myalgic encephalomyelitis/chronic fatigue syndrome (ME/CFS), a chronic disease that can be triggered by multiple factors, particularly infectious diseases [8]. This complex disease is characterised by symptoms such as pathological fatigue and malaise, which can be triggered by minimal physical or cognitive efforts (post-exertional malaise), in addition to unrefreshing sleep, cognitive impairment, orthostatic intolerance, and pain (muscle and/or join pain and headaches). The symptoms provoked by efforts may be experienced immediately or be delayed for hours or days, and the disease leads to a significant reduction in functional activities [9–11]. Despite the significant impact of ME/CFS on the lives of those affected, there is still a scarcity of well-designed and well-powered epidemiological and socio-economic studies in Europe, which could provide reliable estimates on the burden of ME/CFS [12, 13]. Estimation of ME/CFS burden is also made difficult by the lack of biomarkers and by the fact that diagnosis has to be clinical, based on detailed clinical history and physical examination by a clinician who has experience with this disease [10].

In this study, our aim was to evaluate incidence rates of Lyme disease in the UK and assess whether Lyme disease was associated with subsequent (1) fatigue and (2) ME/CFS using a large UK primary care database.

## Materials and methods

### Study design and data source

This is a population-based historical cohort study with a matched comparator cohort using IQVIA Medical Research Data (IMRD), which incorporates data from The Health Improvement Network (THIN), a Cegedim database. It consists of non-identified longitudinal records of approximately 6% of UK primary care patients and it is known to be nationally representative [14, 15]. Patients are informed of the data collection scheme by the practice and have the

ability to opt-out of the database at any time. This study was approved by IMRD Scientific Research Committee and by LSHTM Ethics Committee.

## Study population

Patients who had a first record of Lyme disease between 01 January 2000 and 31 December 2018 and had contributed at least 6 months of high quality data prior to the index date were included in the study. This meant at least 6 months were required to have elapsed between the last date of the 1) adoption of the "Vision" data collection software by the GP; 2) acceptable mortality recording [16]; and 3) patient registration to the GP, and the index date. Patients who had a record of any of the following were excluded: 1) ME/CFS or an underlying chronic condition likely to lead to fatigue any time before the index date; 2) post-viral fatigue, symptoms of fatigue or of any acute conditions likely to lead to fatigue within 12 months before index; or 3) pregnancy within 12 months before index (S1 Table). For each Lyme patient, up to 4 unexposed patients were identified with the same year of birth, sex, and General Practice (GP), provided they met all inclusion and exclusion criteria above and had no prior Lyme disease diagnosis, to form the comparator cohort. When more than 4 patients were available, 4 were randomly sampled. Diagnostic codelists used in this study are available for download from LSHTM Data Compass
(https://doi.org/10.17037/DATA.00002625).

## Index date and follow-up time

The index date was defined in the Lyme cohort as the first record of Lyme disease and in the comparator cohort as the index date of the patient they were matched to. For the Lyme cohort, when more than 12 months elapsed between 2 diagnoses of Lyme disease, participants were considered as infected multiple times [17] and the index date was defined as the date of first infection.

Follow-up period started at the index date and participants were censored at the first occurrence of: death, diagnosis of a comorbidity likely to lead to fatigue, practice deregistration, practice administrative censoring, event of interest, or end of study period (31 January 2020). For the comparator cohort, censoring events additionally included date of Lyme disease diagnosis.

## Exposure

The code list for Lyme disease was adapted from previous studies [18, 19] and included codes for erythema migrans [20]. Incidence analyses relied on patients with codes for suspected and confirmed disease whereas analysis related to the association with fatigue relied only on patients with confirmed codes. Patients with codes for suspected and confirmed disease were indexed on the day the first record of Lyme disease was identified, whether the code corresponded to a suspected or confirmed disease.

## Outcomes

The two main outcome measures were time from index date to (1) consultation for any fatigue-related symptoms or diagnosis (including symptoms of fatigue, post-viral fatigue diagnosis, and ME/CFS) and (2) consultation for a diagnosis of ME/CFS. Code lists were developed based upon clinical terms used for the LSHTM ME/CFS biobank project [21] and were supplemented based on a previous study [22]. A sensitivity analysis was performed including only consultations occurring 6 months or more after the index date with the aim to exclude fatigue

events that could be considered as known symptoms of the ongoing infection, rather than long-term consequences.

## Variables

Covariates of interest were those likely to be associated with the exposure, Lyme disease, and consequently with outdoor activities, and those likely to be associated with fatigue. Covariates considered for this study were based on data availability and included age, body mass index (BMI), smoking status, healthcare utilisation frequency within 6 months prior to the index date, history of depression, index season, and antibiotic treatment at index (S1 Appendix). Any record of amoxicillin, azithromycin, cefotaxime, ceftriaxone and doxycycline within 30 days of index date (and within 30 days after confirmed diagnosis for Lyme cohort patients, if the code for confirmed disease was recorded after the code for suspected disease) were considered. Available covariates were all tested in the final model.

## Missing data

A total of 7,491 participants (70.4%) had complete data: 69.9% of comparators and 72.3% of Lyme patients. In total, 1,753 participants had missing values on smoking status (16.5%), and 3,043 on BMI (28.6%). For variables based on code identification, absence of code was interpreted as the absence of the disease or treatment of interest.

## Statistical analysis

Analyses were conducted using STATA statistical software version 15.1. Associations between Lyme disease and covariates were assessed using Chi-squared tests except for follow-up times of participants, which were compared using Wilcoxon rank sum tests due to a right-skewed distribution. After confirmation that 1) fatigue incidence rates decreased over time and 2) the proportional hazards assumption was valid, Cox regression models were performed and adjusted for the matched variables in accordance with earlier publications [23, 24], except for ME/CFS, which was a rare outcome. This approach gave the ability to simultaneously account for the effects of age, sex, as well as socio-economic status and diagnosis practices (via the adjustement by GP). Since the analytic approach was based on a complete case analysis, variables with no clear confounding, modifying, or risk factor effect in this study were excluded from the adjusted model to avoid reducing further the size of the population and thus the study power. Robust standard errors were used to adjust for clustering by the unique patient identifier variable [24] to account for the fact that 28 out of the 2130 participants of the Lyme cohort (1.3%) also contributed to the comparator cohort. For the 6-months sensitivity analysis, patients followed-up for less than 6 months were excluded, as well as comparators matched to a Lyme patient followed for less than 6 months.

## Results

### Incidence rates

Among the 4,973 patients with a Lyme disease diagnosis during the study period, 4,947 had no diagnosis prior to the study period and were considered as incident cases. Incidence rate for Lyme disease over the study period was 5.18 per 100,000 py (Table 1). Incidence rates were similar in males and females (5.08 and 5.28 per 100,000 py, respectively). Rates were highest in 55–64 years old (7.25 per 100,000 py), contrasting with rather low rates in children under 15 (3.71 per 100,000 py) and in individuals above 75 (2.20 per 100,000 py). The youngest patient with Lyme disease was 7 months at diagnosis and the oldest was 94 years old. Incidence rate in

**Table 1. Incidence rates of Lyme disease per sex, age, geography, and time period[a].**

| | Number of cases | Population at risk in million person-years | Incidence rates per 100,000 person-years |
|---|---|---|---|
| **Total** | 4,947 | 95.5 | 5.18 |
| **Sex** | | | |
| Female | 2,536 | 48.0 | 5.28 |
| Male | 2,411 | 47.5 | 5.08 |
| **Age in years** | | | |
| <15 | 576 | 15.5 | 3.71 |
| 15–24 | 485 | 10.7 | 4.54 |
| 25–34 | 673 | 12.9 | 5.20 |
| 34–44 | 834 | 14.4 | 5.79 |
| 45–54 | 844 | 13.7 | 6.15 |
| 55–64 | 845 | 11.7 | 7.25 |
| 65–74 | 518 | 8.8 | 5.91 |
| > = 75 | 172 | 7.8 | 2.20 |
| **Country** | | | |
| England | 2,358 | 67.1 | 3.51 |
| Wales | 278 | 10.8 | 2.57 |
| Scotland | 2,218 | 14.5 | 15.32 |
| Northern Ireland | 93 | 3.1 | 2.96 |
| **Season[b]** | | | |
| January to March | 515 | 23.9 | 2.16 |
| April to June | 1,059 | 23.9 | 4.44 |
| July to September | 2,351 | 23.9 | 9.85 |
| October to December | 1,022 | 23.9 | 4.28 |
| **Years covered** | | | |
| 2000 to 2004 | 783 | 24.2 | 3.15 |
| 2005 to 2009 | 1,496 | 28.4 | 5.13 |
| 2010 to 2014 | 1,548 | 27.7 | 5.44 |
| 2015 to 2018 | 1,120 | 15.2 | 7.20 |

[a]All rows cover the study period (01Jan2000 to 31Dec2018), except for "Years covered" where the period of interest is indicated on each row; Number of person-years at risk were calculated as the sum of active patients in the database on the 1st of July for all years included.

[b]For the break-down, the total number of person-years was calculated as 1/4 of the number of person-years for each year of the study period and is thus the same for all 4 seasons.

Scotland reached 15.32 per 100,000 py over the study period, approximately 5 times higher than in England, Wales, and Northern Ireland. Expectedly, summer was the season when most diagnoses were made (incidence rate of 9.85 per 100,000 py), whereas winter was much quieter (2.16 per 100,000 py). Overall, there was a clear trend for increase of incidence rates over the study period from 3.15 per 100,000 py in 2000–2004 to 7.20 in 2015–2018, cumulating in a rate of 9.33 rate in 2018 (Fig 1). This overall increase over time was observed in all countries except Scotland, in which incidence rates grew steadily until 2010 but then fluctuated from 2010 and 2018.

## Participants characteristics

In total, 2,130 patients were eligible for the analysis (Fig 2). The comparator cohort consisted of 8,510 patients, i.e. a Lyme: non-Lyme ratio of 1:4 for 2,121 Lyme patients, 1:3 for 8 Lyme patients and 1:2 for 1 Lyme patient. The majority of patients (55.3%) were aged between 35

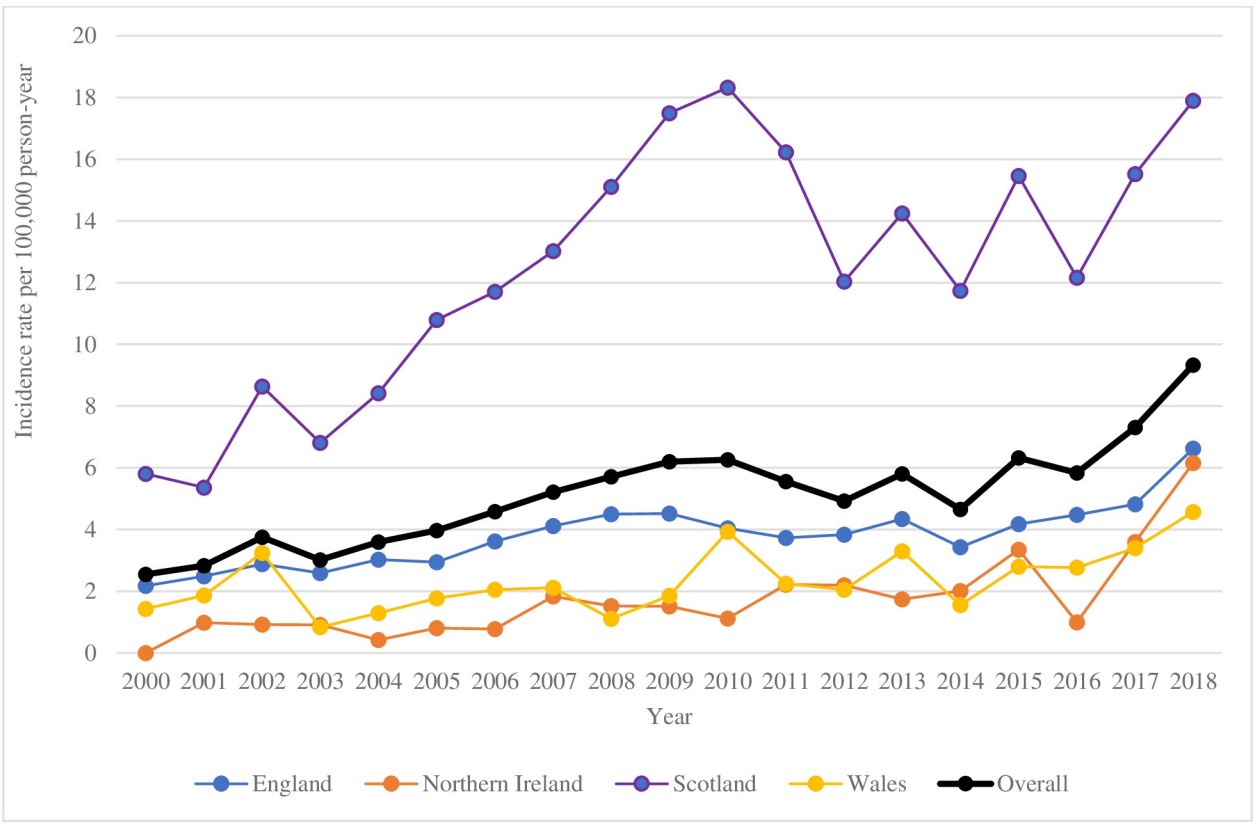

**Fig 1. Evolution of Lyme disease incidence rate over study period per country.**

and 64 years old (Table 2). The Lyme cohort had a higher proportion of healthy weight persons (45.7% versus 40.5% in comparators) and of never smokers (59.5% versus 54.2%). They tended to consult their GPs more often (21.8% had no consultation in the last 6 months, versus 38.2% in comparators).

Antibiotic treatment was prescribed for the vast majority of Lyme patients (73.1%). The most prescribed antibiotic was doxycycline, as unique treatment in 1,228 patients (57.7%), with amoxicillin in 49 patients (2.3%), or with azithromycin in 2 patients (0.1%). Amoxicillin was also commonly prescribed as a unique treatment in 264 patients (12.4%).

Nearly all Lyme disease patients had the most common form of the disease Lyme borreliosis (2,108 out of 2,130); 5 had Lyme neuroborreliosis and 17 had Lyme arthritis. Follow-up time was similar in Lyme and comparator cohorts, with a median of 3.4 and 3.7 years, respectively, for any types of fatigue and 4.1 years both in Lyme and comparator cohorts for ME/CFS.

## Crude association between Lyme disease and fatigue

With 929 events, overall incidence of any types of fatigue reached 192.62 for 10,000 py with a median incidence time of 2.3 years after index. Incidence rate for ME/CFS was much lower with 3.29 per 10,000 py, i.e. 17 participants experiencing an event. Incidence rates were 165.60 per 10,000 person-years in unexposed comparators and 307.90 in Lyme disease cases for any types of fatigue (Table 3), i.e. a crude HR of 2.27 (95% CI: 1.95–2.64). A striking difference was observed for ME/CFS: 1.20 per 10,000 person-years in comparators and 11.76 in Lyme disease cases, i.e. a crude HR of 9.76 (95%CI: 3.44–27.70).

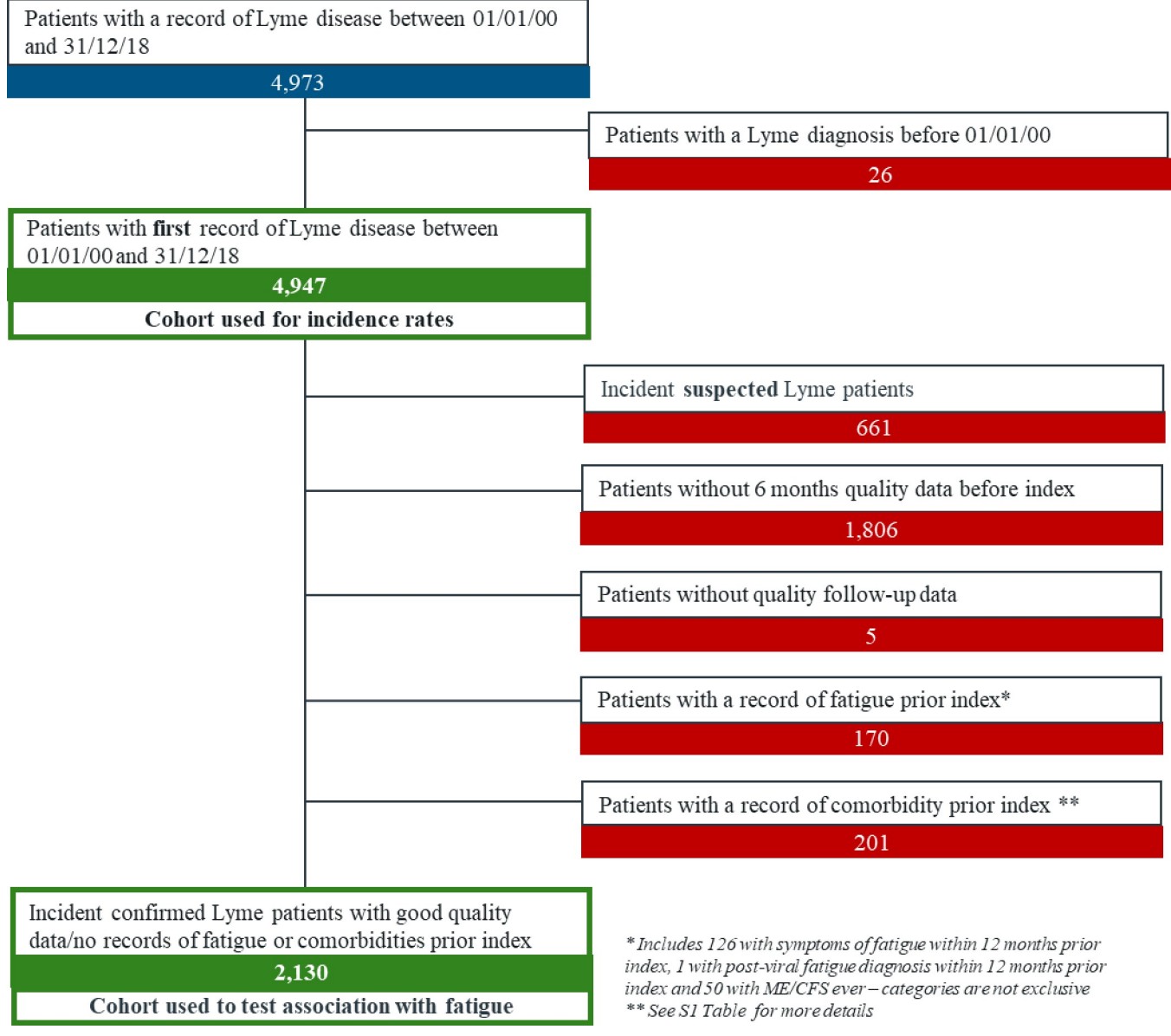

**Fig 2. Flowchart of patients included in the Lyme cohort.**

## Crude association between covariates and fatigue

Most covariates were associated with any types of fatigue in crude analyses (S2 Table): obese participants, participants with a history of depression and those who were prescribed an antibiotic treatment at time of index were more likely to have records of fatigue. Healthcare utilisation frequency appeared to be strongly correlated with any types of fatigue with a linear effect; patients who visited their GP more than 10 times in the 6 months prior to the index date had more than 7 times the risk of suffering from fatigue.

Associations of covariates were difficult to interpret for ME/CFS as 95% confidence intervals were large. However, the associations of history of depression and antibiotic treatment at the time of index appeared to be even stronger than for any types of fatigue. Despite low

**Table 2. Characteristics of study participants at index date.**

| | Lyme cohort[a] | Comparator cohort (Non-Lyme) |
|---|---|---|
| | N = 2,130 | N = 8,510 |
| | n (%) | n (%) |
| **Sex** | | |
| Female | 1,034 (48.5) | 4,132 (48.6) |
| Male | 1,096 (51.5) | 4,378 (51.4) |
| **Age at diagnosis, in years** | | |
| <15 | 268 (12.6) | 1,072 (12.6) |
| 15–24 | 150 (7.0) | 600 (7.1) |
| 25–34 | 206 (9.7) | 824 (9.7) |
| 34–44 | 352 (16.5) | 1,407 (16.5) |
| 45–54 | 398 (18.7) | 1,590 (18.7) |
| 55–64 | 427 (20.1) | 1,708 (20.1) |
| 65–74 | 251 (11.8) | 1,003 (11.8) |
| > = 75 | 78 (3.7) | 306 (3.6) |
| **Practice country** | | |
| England | 1,128 (53.0) | 4,511 (53.0) |
| Northern Ireland | 38 (1.8) | 152 (1.8) |
| Wales | 87 (4.1) | 347 (4.1) |
| Scotland | 877 (41.1) | 3,500 (41.1) |
| **Body Mass Index[b,c]** | | |
| Underweight | 32 (2.1) | 140 (2.3) |
| Healthy weight | 710 (45.7) | 2,448 (40.5) |
| Overweight | 545 (35.1) | 2,131 (35.3) |
| Obese | 266 (17.1) | 1,325 (21.9) |
| *Unknown* | *577* | *2,466* |
| **Smoking status[c]** | | |
| Never smoker | 1,079 (59.5) | 3,833 (54.2) |
| Ex-smoker | 443 (24.4) | 1,570 (22.2) |
| Current smoker | 293 (16.1) | 1,669 (23.6) |
| *Unknown* | *315* | *1,438* |
| **Healthcare utilisation frequency[d]** | | |
| 0 | 465 (21.8) | 3,253 (38.2) |
| 1 | 356 (16.7) | 1,639 (19.3) |
| 2 to 4 | 702 (33.0) | 2,275 (26.7) |
| 5 to 9 | 440 (20.7) | 1,026 (12.1) |
| >10 | 167 (7.8) | 317 (3.7) |
| **History of depression[e]** | | |
| Yes | 369 (17.3) | 1,493 (17.5) |
| **Antibiotic treatment[f]** | | |
| Yes | 1,556 (73.1) | 201 (2.4) |
| **Index season** | | |
| January to March | 171 (8.0) | 684 (8.0) |
| April to June | 470 (22.1) | 1,877 (22.1) |
| July to September | 1,043 (40.0) | 4,165 (48.9) |
| October to December | 446 (21.0) | 1,784 (21.0) |
| **Follow-up times in years** | | |
| Median (min-max) for any types of fatigue | 3.4 (0–18.6) | 3.7 (0–19.5) |

(*Continued*)

Table 2. (Continued)

| | Lyme cohort[a] | Comparator cohort (Non-Lyme) |
|---|---|---|
| | N = 2,130 | N = 8,510 |
| | n (%) | n (%) |
| Median (min-max) for ME/CFS | 4.1 (0–18.6) | 4.1 (0–19.5) |

[a]Patients with confirmed diagnosis only

[b]Underweight<18.5; healthy weight:18.50–24.99; overweight 25–29.99; obese≥ 30.00, in kg/m$^2$

[c]For parameters with missing data, percentages are calculated using number of observations with complete data as denominator.

[d]Number of GP encounters within 6 months prior index

[e]Any record of mild or moderate depression prior index

[f]Record of antibiotics (amoxicillin, azithromycin, cefotaxime, ceftriaxone, doxycycline) used against Lyme disease within 30 days of index date or within 30 days after confirmed diagnosis for Lyme cohort patients, if different.

numbers in each category, there was evidence that the highest category of healthcare utilisation frequency was strongly associated with diagnosis of ME/CFS.

## Adjusted association between Lyme disease and fatigue

Cox regression for any types of fatigue was adjusted for healthcare utilisation frequency, antibiotic treatment, and season of index in addition to age, sex, and GP via matching. In autumn and winter, the association between Lyme disease and any types of fatigue was strongest with an HR of 3.14 (95%CI: 1.92–5.13; p<0.001) and 2.23 (95%CI: 1.21–4.11; p = 0,010), respectively (Fig 3). For other seasons the effect was more subtle: 1.32 (95%CI: 0.83–2.10; p = 0,247) in spring and 1.52 (95%CI: 1.03–2.24; p = 0.034) in summer. The sensitivity analysis on fatigue events recorded more than 6 months after index showed an attenuated, but still present effect of Lyme disease on any types of fatigue with adjusted HR of 2.15 (95%CI: 1.19–3.87) in winter, 1.07 (95%CI: 0.67–1.70) in spring, 1.22 (95%CI: 0.81–1.83) in summer, and 2.27 (95%CI: 1.36–3.78) in autumn.

Challenged by the fact that the low number of events allowed the addition of only 1 or 2 parameters to the model studying ME/CFS on its own, only the strongest confounder, antibiotic treatment, was used to adjust the final model. The effect of Lyme disease on ME/CFS appeared to be strong with an adjusted HR for antibiotic treatment of 16.95 (95%CI: 5.17–55.60) in the main analysis and 8.29 (95%CI: 2.13–32.22) 6 months after index.

## Discussion

### Comparison to previous studies

The present study reports an incidence rate of Lyme disease between 2000 and 2018 in the UK of 5.18 per 100,000 py. This is higher than the rates of 0.53 for 2015 assessed from an administrative hospital dataset [19] and of 2.70 calculated from number of cases shared by Public Health England in 2017 in England and Wales. The estimate is however lower than the rate of 12.1 obtained with a comparable study design and a similar data source [18]. This difference in rates likely reflects the difference in case definition; Cairns et al. almost doubled the study population by including patients with a mention of Lyme disease or erythema migrans in medical notes or with a record of a Lyme test lacking results, provided a simultaneous record of antibiotic prescription was found, whereas in the current study we only took into consideration firm diagnoses translated into read codes and positive Lyme test results.

**Table 3. Incidence rates and crude hazard ratios (HR) of any types of fatigue and ME/CFS for Lyme disease.**

| | Participants with outcomes | Total person-years | Incidence rates[a] | Crude HR[b] (95% CI) |
|---|---|---|---|---|
| **Any types of fatigue** | | | | |
| *Straight after index[c]* | | | | |
| Overall | 929 | 48,228 | 192.62 | N.A. |
| Comparator cohort (non-Lyme) | 647 | 39,070 | 165.60 | ref |
| Lyme cohort | 282 | 9,159 | 307.90 | 2.27 (1.95–2.64) |
| *6 months after index[d]* | | | | |
| Overall | 802 | 42,148 | 190.28 | N.A. |
| Comparator cohort (non-Lyme) | 573 | 33,767 | 169.69 | ref |
| Lyme cohort | 229 | 8,381 | 273.24 | 1.80 (1.53–2.12) |
| **ME/CFS** | | | | |
| *Straight after index[c]* | | | | |
| Overall | 17 | 51,700 | 3.29 | N.A. |
| Comparator cohort (non-Lyme) | 5 | 41,495 | 1.20 | ref |
| Lyme cohort | 12 | 10,205 | 11.76 | 9.76 (3.44–27.70) |
| *6 months after index[d]* | | | | |
| Overall | 14 | 45,073 | 3.10 | N.A. |
| Comparator cohort (non-Lyme) | 5 | 35,861 | 1.39 | ref |
| Lyme cohort | 9 | 9,212 | 9.77 | 7.02 (2.35–20.94) |

[a] Incidence per 10,000 person-years

[b] using Cox regression with adjustment for the match variable (i.e. age, sex, and general practice) for any types of fatigue and no adjustement for ME/CFS

[c] consultations occurring any time after the index date

[d] only consultations occurring 6 months or more after the index date, with 1,987 Lyme patients and 7,577 comparators.

The present results show an approximately 2- and 3-fold increase in any subsequent fatigue in winter and autumn, respectively, and a 16-fold increase in ME/CFS (all seasons combined). Association between Lyme disease and fatigue is in agreement with a cohort study assessing fatigue severity (any types) 30 months after treatment in Norway [7]. It contrasts, however, with other studies that did not detect any effect [4–6], which could be due, at least for the 2 earliest, to their small sample sizes (less than 100 Lyme patients). Unfortunately, the recent systematic review on Lyme disease effect on overall symptoms did not include a separate analysis for fatigue that could have enabled a comparison with the present results [25]. Of note, they observed that the association between Lyme disease and overall symptoms was usually attenuated when adding possible cases, which was not the case in the present study (S3 Table).

## Strengths

With more than 2000 Lyme patients, this study is one of the largest so far to examine the association between Lyme disease and subsequent fatigue. The fact that participation in the IMRD database relies on an opt-out system reduced non-response rates and consequently the risk of selection bias.

The results presented highlight the importance of diagnosis season in the association between Lyme disease and any subsequent fatigue: association was stronger for patients whose infection was reported in autumn or in winter, compared to spring or summer. The data used did not allow to assess date of infection for most patients, given their date of diagnosis could have taken place weeks or months after the infection. This holds true for all diagnosis codes, except for erythema migrans, which is detected shortly after infection. Interestingly, in a post-

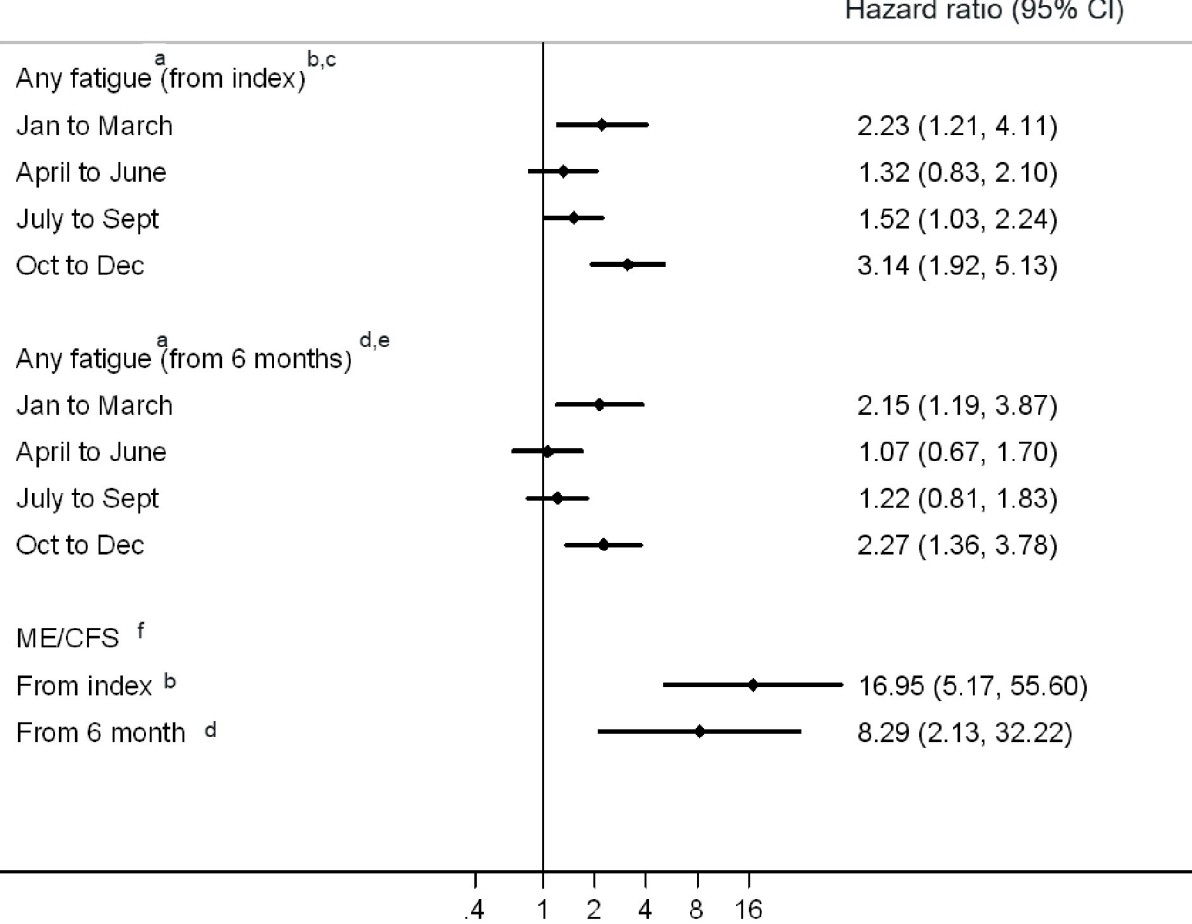

Fig 3. Hazard ratios for incidence of any fatigue and chronic fatigue syndrome in Lyme disease patients versus comparator (non-Lyme disease) patients.

[a] Using Cox regression with robust standard errors, adjustment for the match variable (i.e. age, sex, general practice), for healthcare utilisation frequency as a continuous variable, prescribed antibiotic treatment, and with index season as effect modifier, [b] consultations occurring any time after the index date, [c] p value for interaction as per likelihood rate ratio test (LRT): 0.008, [d] only consultations occurring 6 months or more after the index date, with 1,987 Lyme patients and 7,577 comparators, [e] p value for interaction as per LRT: 0.060, [f] using Cox regression with robust standard errors and with adjustment for prescribed antibiotic treatment

**Fig 3. Hazard ratios for incidence of any fatigue and chronic fatigue syndrome in Lyme disease patients versus comparator (non-Lyme disease) patients.**

hoc analysis looking at the association between Lyme disease and any types of fatigue in patients who had an erythema migrans code, i.e. in patients for whom the index date truly reflects infection date, there was an attenuation of the differences between seasons (p value 0.3490 compared to 0.0008 for the confirmed cohort). This suggests that the difference in association between seasons could, at least in part, reflect the delay in diagnosis, and thus the delay in treating patients.

## Limitations

Firstly, we cannot exclude misclassification of the exposure for several reasons: 1) Patients whose diagnosis was not entered as a formal read code but added instead as comment in medical notes

would not have been included. 2) Patients with a diagnosis of Lyme disease made outside the GP–for instance at the hospital—would not necessarily have this information fed back into the GP records, especially if the initial course of antibiotics was not extended. 3) Some individuals with Lyme disease might not have sought medical advice and thus might have remained undiagnosed. 4) Inversely, due to the complexity of the disease, many patients diagnosed as Lyme disease might suffer in fact from another disease [26, 27]. It is expected that these misclassifications would be non-differential, which would have diluted the strength of the association.

Secondly, recording and diagnosis of fatigue-related symptoms and conditions is expected to differ from one practitioner to another, which might lead to misclassification of the outcome, which is why we matched by GP. This approach would have accounted for the known underdiagnosis of ME/CFS by some GPs who still do not believe this disease exists or do not understand it well enough to make the diagnosis. This approach, however, would not prevent the possibility of differential misclassification of the outcome occurring when GPs decide not to record diagnosis of fatigue when they know their patient has Lyme disease. This differential misclassification is even more likely for ME/CFS, whose diagnosis usually comes after excluding anything else that might be causing the fatigue. For instance, we cannot exclude the possibility that some patients with ME/CFS have no record of this disease because they were misdiagnosed as having Post-Treatment Lyme Disease Syndrome (PTLDS), a term used to describe patients with *Borrelia burgdorferi* infection who report persisting symptoms despite adequate antibiotic treatment [28]. These two syndromes share many features indeed [29], although recent findings on cerebrospinal fluid proteomes suggest differentiating diagnostic tools could be developed [30]. A differential misclassification of ME/CFS would have lead to a dilution of the observed effect. In contrast, it is possible that patients diagnosed with Lyme disease are more attentive to their symptoms and more likely to visit their GP when experiencing signs of fatigue. In a survey sponsored by the Centers for Disease Control and Prevention in the United States, 0.5% of respondents reported having chronic Lyme disease [31]. The disease is better known in the US where incidence rates are higher. However, awareness in the UK has grown in past years and national experts have underlined the anxiety of the public around Lyme disease and the role played by the media in fueling this anxiety when describing chronic Lyme disease [27]. In that sense, it might be difficult to disentangle the effect of the disease on triggering the "feeling" of fatigue from the effect of triggering fatigue itself. The association of Lyme disease and more severe forms of fatigue like ME/CFS should probably be considered from a different perspective given history of infections was reported as a strong risk factor for ME/CFS [32]. The likely explanation is that infections trigger host responses in predisposed individuals that lead to prolonged symptoms and chronic fatigue [8].

Thirdly, it is possible that results suffer from residual confounding. For instance, patients with Lyme disease might have been infected with other known or unknown diseases transmitted by ticks at the time of the bite, and these diseases might have led to long-term fatigue. It is also possible that there is residual confounding on the variables we have integrated in the model: for instance, we found that 27.0% of patients diagnosed with Lyme disease had no record for treatment with antibiotics. We might thus have misclassified this confounder in patients who received a prescription from a consultant not reported to the GP. Finally, whilst the database enabled us to identify patients who received a prescription of antibiotics, we cannot determine their adherence to the prescribed treatment.

### Implications and future research

Based on this study, the extrapolated number of Lyme disease cases in the UK would have been approximatively 3,200 per year between 2000 and 2018 on average, with a peak of 6,500

in 2018 using mid-year population estimates from the Office for National Statistics over the study period.

Initial analyses 6 months after index show that among the 1,945 patients infected only once with bacterium *Borrelia burgdorferi*, 218 (11.2%) had a record for fatigue of any types and 9 (0.5%) had a record of ME/CFS. An interesting aspect to explore in future research concerns Lyme patients with multiple infections: 10 out of 37 patients with 2 infections (27.0%), none of 4 with 3 infections, and 1 out of 1 with 4 infections had fatigue symptoms, suggesting a potential association between number of infections and incidence of fatigue of any types ($p<0.001$). However, this association was not observed for ME/CFS since none of the Lyme patients suffering from ME/CFS were infected several times (S4 Table).

Given the low incidence of ME/CFS, the use of large, routinely collected data is appropriate for studying the association between Lyme disease and ME/CFS. However, validation of diagnoses of ME/CFS made in GP practices using specific questionnaires would enable us to understand the extent to which GP databases in the UK are reliable in terms of recording ME/CFS, an information which might also prove useful when assessing the association of ME/CFS with viral infections such as COVID-19.

## Conclusions

The present study reports an increased incidence rate of Lyme disease over time in the UK. It also shows an association between Lyme disease and fatigue of any types, an association that weakens but is still clearly present after 6 months, and varies with season. Large studies are needed to confirm the association with ME/CFS. These results support NICE recommendations for assessment and management of symptoms amongs patients previously treated with Lyme disease, including chronic pain, depression and anxiety, fatigue, and sleep disturbance [20].

## Supporting information

**S1 Table. List of comorbidities.**
(DOCX)

**S2 Table. Incidence rates and hazard ratios of any types of fatigue and of chronic fatigue syndrome for study covariates.**
(DOCX)

**S3 Table. Incidence rates, crude and adjusted hazard ratios of any types of fatigue and ME/CFS for Lyme disease using an extended cohort including also suspected Lyme disease patients and their matched controls.**
(DOCX)

**S4 Table. Incidence of any types of fatigue and ME/CFS in patients with multiple infections with bacterium *Borrelia burgdorferi*.**
(DOCX)

**S1 Appendix. Description and categorization of covariates.**
(DOCX)

## Acknowledgments

The authors would like to thank Bassam Bafadhal, Itohan Evbuomwan, and Louise Pinder, IQVIA London, for helpful advice on IMRD.

## Author Contributions

**Conceptualization:** Florence Brellier, Kevin Wing.

**Data curation:** Florence Brellier.

**Formal analysis:** Florence Brellier.

**Investigation:** Florence Brellier, Mar Pujades-Rodriguez, Kevin Wing.

**Methodology:** Florence Brellier, Mar Pujades-Rodriguez, Emma Powell, Kathleen Mudie, Eliana Mattos Lacerda, Luis Nacul, Kevin Wing.

**Project administration:** Florence Brellier.

**Resources:** Florence Brellier.

**Software:** Florence Brellier.

**Supervision:** Florence Brellier, Kevin Wing.

**Validation:** Florence Brellier, Kevin Wing.

**Visualization:** Florence Brellier, Kevin Wing.

**Writing – original draft:** Florence Brellier.

**Writing – review & editing:** Florence Brellier, Mar Pujades-Rodriguez, Emma Powell, Kathleen Mudie, Eliana Mattos Lacerda, Luis Nacul, Kevin Wing.

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
