## [Decision Letter · Decision Letter 0]

15 Oct 2021

PONE-D-21-25917Incidence of Lyme disease in the United Kingdom and association with fatigue: a population-based, historical cohort studyPLOS ONE

Dear Dr. Brellier,

Thank you for submitting your manuscript to PLOS ONE. After careful consideration, we feel that it has merit but does not fully meet PLOS ONE’s publication criteria as it currently stands. Therefore, we invite you to submit a revised version of the manuscript that addresses the points raised during the review process.

We look forward to receiving your revised manuscript.

Kind regards,

Utpal Pal, PhD

Academic Editor

PLOS ONE

Journal Requirements:

2. In ethics statement in the manuscript and in the online submission form, please provide additional information about the patient records/samples used in your retrospective study. Specifically, please ensure that you have discussed whether all data/samples were fully anonymized before you accessed them and/or whether the IRB or ethics committee waived the requirement for informed consent. If patients provided informed written consent to have data/samples from their medical records used in research, please include this information.

5. Please upload a new copy of Figure 2 as the detail is not clear. Please follow the link for more information: https://blogs.plos.org/plos/2019/06/looking-good-tips-for-creating-your-plos-figures-graphics/" https://blogs.plos.org/plos/2019/06/looking-good-tips-for-creating-your-plos-figures-graphics/

6.Please review your reference list to ensure that it is complete and correct. If you have cited papers that have been retracted, please include the rationale for doing so in the manuscript text, or remove these references and replace them with relevant current references. Any changes to the reference list should be mentioned in the rebuttal letter that accompanies your revised manuscript. If you need to cite a retracted article, indicate the article’s retracted status in the References list and also include a citation and full reference for the retraction notice.

Reviewers' comments:

Reviewer's Responses to Questions

**Comments to the Author**

1. Is the manuscript technically sound, and do the data support the conclusions?

Reviewer #1: Yes

Reviewer #2: Yes

2. Has the statistical analysis been performed appropriately and rigorously? 

Reviewer #1: Yes

Reviewer #2: Yes

3. Have the authors made all data underlying the findings in their manuscript fully available?

Reviewer #1: Yes

Reviewer #2: Yes

4. Is the manuscript presented in an intelligible fashion and written in standard English?

Reviewer #1: Yes

Reviewer #2: Yes

5. Review Comments to the Author

Reviewer #1: In the current manuscript, authors have conducted a survey on Lyme disease patients, diagnosed during 2000-2018, and tried to find out its possible association with long-term fatigue. The authors have used large number of datasets for analysis, which suggested the strong association between Lyme disease and fatigue or ME/CSF. Overall, the current study provides new insights about Lyme disease and it will be useful for future Lyme disease guideline management. However, there are few minor points which should be addressed-

1. Incorporate information regarding Post-Treatment Lyme Disease Syndrome (PTLDS) in discussion section as it is often misdiagnosed with chronic fatigue syndrome.

2. Add more information about Chronic fatigue syndrome as it is very complex condition which can be triggered by combination of factors. Even there is no single test available to confirm chronic fatigue syndrome.

3. The quality of figure1and 2 is poor, please improve it. Figure 2 is not even readable.

4. Please italicize the Bacterium name mentioned throughout the manuscript.

Reviewer #2: The manuscript entitled “Incidence of Lyme disease in the United Kingdom and association with fatigue: a population-based, historical cohort study” with the number PONE-D-21-25917 has been evaluated. It is a retrospective cohort study. The authors determined the incidence of Lyme disease (LD) and the effect of LD on fatigue symptoms using the accumulated data of a large population in UK, between 2000-2018.

In the manuscript, 2,130 patients with a diagnosis of Lyme disease were investigated in comparison with the control group of non-Lyme disease population constituted of 4 times the quantity of Lyme patients. The targeted Lyme disease patients were selected based on appropriate criteria. The control group population was selected based on the similar features that of Lyme disease patients. The study population is sufficient in terms of the quantity and the features for statistical analysis. These are a few of the positive aspects of the investigation. In addition, the association of Lyme disease with fatigue-related symptoms and diagnose of myalgic encephalomyelitis/chronic fatigue syndrome (ME/CFS) were investigated at least for 3 years after Lyme disease diagnosis. The effect of various variables such as age, sex, season, obesity and antibiotic therapy were evaluated in terms of fatigue cases determined in Lyme disease patients. Antibiotic therapy was found as an important covariable that would cause fatique in the treated Lyme patients. Depending on the frequency of the doctor visit of Lyme patients, fatique was increasingly detected in the patients that visited the clinics more often. Therefore, healthcare utilisation frequency was found as one of the major variables. Seasonal changes were also found to be important for fatique symptoms. Effect of Lyme disease on fatique was demonstrated as higher during autumn and winter.

In this manuscript, demonstration of the persistence of the fatigue symptoms and also ME/CFS symptoms for more than 6 months (though decreased) following Lyme disease diagnosis are interesting findings. Evaluation of the data of a large number of Lyme disease patients in comparison with higher number of comparators increased reliability.

6. PLOS authors have the option to publish the peer review history of their article (what does this mean?). If published, this will include your full peer review and any attached files.

Reviewer #1: No

Reviewer #2: No

---

## [Author Response · Author response to Decision Letter 0]

22 Feb 2022

Feb22:

Dear editor,

I have received the following request: "Before we can proceed, please confirm that the authors did not have any special access privileges that others would not have and that others can apply to have access the data in the same manner as the authors."

This seems to be in response to my clarification of the data access procedure shared on 24 November 2021, and confirmed by me on 10 February 2022.

I am struggling to understand why editors would like me to add such a sentence in the data access section of my manuscript. Access to patient-level health data is extremely regulated in Europe and most of the time data owners do not accept/are not allowed to share their data. IMRD is luckily accessible to all and I have already described in the data access section how readers could proceed if they'd like to ask for access. Now I did access these data while employed by IQVIA who already has a licence agreement to access IMRD data. It is such possible that I benefitted from shorter timelines because of this. I thus do not want to state that external readers would have exactly the same access privileges. However you must have noted the current sentence in my statement: "Researchers have the possibility to access IMRD similarly to the authors, subject to a sublicense and an approved protocol.", which, I think, addresses your concerns with words I find more appropriate.

I would like to suggest please that you give me a phone call on *[phone number redacted]* if you'd like to discuss this further. As I said, I have answered your first question on data access almost 3 months ago ( see pdf) and it is only now that I am being asked for clarifications.

I am looking forward to hearing from you shortly to finalize these quality checks as soon as possible and enable the reviewers to review the manuscript.

Thank you in advance,

Florence Brellier

Feb 10:

Hello,

I received recently a request from the journal to confirm the Data Availability statement for my manuscript: as per my previous communications, I propose the following:

"The data underlying the results presented in the study are from IQVIA Medical Research Data (IMRD), which incorporates data from The Health Improvement Network (THIN), a Cegedim database. Authors do not have the permission to share the data. Researchers have the possibility to access IMRD similarly to the authors, subject to a sublicense and an approved protocol. They may contact IMRDEnquiries@iqvia.com for this purpose."

Thank you in advance and kind regards,

Florence Brellier

[17 Dec] Dear editor,

We have been asked to answer questions related to Competing Interest statement. 

Please note that there seem to have been some misunderstanding on the information we shared: Dr. Pujades-Rodriguez was not an employee at Union Chimique Belge (UCB) Biopharma before the start of the study, she became an employee at Union Chimique Belge (UCB) Biopharma after the study was finalized. Same actually applies to Dr. Brellier who is now an employee at Bristol Myers Squibb but started this employment after the study was finalized.

Thus the answer to your questions are the following:

a.) Are there any patents, products in development or marketed products associated with this research to declare in relation to Dr. Pujades-Rodriguez employment with UCB Biopharma? No, and same holds true for Dr. Brellier with Bristol-Myers Squibb

b.) Does Dr. Pujades-Rodriguez employment with UCB Biopharma alter your adherence to PLOS ONE policies on sharing data and materials? No, and same holds true for Dr. Brellier with Bristol-Myers Squibb

In conclusion we have amended the proposed statement to: 

"The authors have read the journal’s policy and have the following competing interests: Mar Pujades-Rodriguez and Florence Brellier are now employees at Union Chimique Belge (UCB) Biopharma and at Bristol Myers Squibb, respectively. They were both IQVIA employees while working on this study and there are no patents, products in development or marketed products associated with this research to declare. This does not alter authors’ adherence to PLOS ONE policies on sharing data and materials."

We hope it clarifies your questions.

We are looking forward to hearing from you,

With kind regards,

Florence Brellier

[24 Nov] Dear editor,

We have been asked to clarify the data access model for other researchers who wish to access the same data as the authors. Please see our response below:

Authors do not have the permission to share the project data.

Researchers have the possibility to access IMRD similarly to the authors, subject to a sublicense and an approved protocol. They may contact IMRDEnquiries@iqvia.com for this purpose.

We hope it clarifies your questions.

We are looking forward to hearing from you,

With kind regards,

Florence Brellier

[10 Nov] Review Comments to the Author

Reviewer #1: In the current manuscript, authors have conducted a survey on Lyme disease patients, diagnosed during 2000-2018, and tried to find out its possible association with long-term fatigue. The authors have used large number of datasets for analysis, which suggested the strong association between Lyme disease and fatigue or ME/CSF. Overall, the current study provides new insights about Lyme disease and it will be useful for future Lyme disease guideline management. However, there are few minor points which should be addressed-

1. Incorporate information regarding Post-Treatment Lyme Disease Syndrome (PTLDS) in discussion section as it is often misdiagnosed with chronic fatigue syndrome.

Reply: we have added a paragraph in the discussion to bring to the reader’s attention this possible misdiagnosis and have added the following references: Nemeth et al., 2016; Gaudino et al., 1997; Schutzer et al. 2011 (see full references below).

2. Add more information about Chronic fatigue syndrome as it is very complex condition which can be triggered by combination of factors. Even there is no single test available to confirm chronic fatigue syndrome.

Reply: We have added a paragraph in the introduction section to give more information on Chronic Fatigue Syndrome supported by the following additional references: Estévez-López et al, 2020; Pheby et al, 2020; Carruthers et al., 2011 ; Institute of Medicine (IOM), 2015 (see full references below).

3. The quality of figure1and 2 is poor, please improve it. Figure 2 is not even readable.

Reply: The Preflight Analysis and Conversion Engine (PACE) digital diagnostic tool has been used to generate a new copy of all figures and these high-resolution figures have now been uploaded. 

4. Please italicize the Bacterium name mentioned throughout the manuscript.

Reply: The text Borrelia burgdorferi has been italicized throughout the manuscript as requested

Reviewer #2: The manuscript entitled “Incidence of Lyme disease in the United Kingdom and association with fatigue: a population-based, historical cohort study” with the number PONE-D-21-25917 has been evaluated. It is a retrospective cohort study. The authors determined the incidence of Lyme disease (LD) and the effect of LD on fatigue symptoms using the accumulated data of a large population in UK, between 2000-2018.

In the manuscript, 2,130 patients with a diagnosis of Lyme disease were investigated in comparison with the control group of non-Lyme disease population constituted of 4 times the quantity of Lyme patients. The targeted Lyme disease patients were selected based on appropriate criteria. The control group population was selected based on the similar features that of Lyme disease patients. The study population is sufficient in terms of the quantity and the features for statistical analysis. These are a few of the positive aspects of the investigation. In addition, the association of Lyme disease with fatigue-related symptoms and diagnose of myalgic encephalomyelitis/chronic fatigue syndrome (ME/CFS) were investigated at least for 3 years after Lyme disease diagnosis. The effect of various variables such as age, sex, season, obesity and antibiotic therapy were evaluated in terms of fatigue cases determined in Lyme disease patients. Antibiotic therapy was found as an important covariable that would cause fatigue in the treated Lyme patients. Depending on the frequency of the doctor visit of Lyme patients, fatigue was increasingly detected in the patients that visited the clinics more often. Therefore, healthcare utilisation frequency was found as one of the major variables. Seasonal changes were also found to be important for fatigue symptoms. Effect of Lyme disease on fatigue was demonstrated as higher during autumn and winter.

In this manuscript, demonstration of the persistence of the fatigue symptoms and also ME/CFS symptoms for more than 6 months (though decreased) following Lyme disease diagnosis are interesting findings. Evaluation of the data of a large number of Lyme disease patients in comparison with higher number of comparators increased reliability.

Reply: Thank you for your comments – no point seemed to require action from our side.

Response to overall Comments 

Reply: The manuscript has been revised according to the style templates, with main changes including updates of affiliations and addition of corresponding author, additions of level1-2-3 headings, figure captions, and references to supplementary information. Please note that IQVIA, the affiliation from the first and second authors, doesn’t have a full form as such [“I” was initially taken from IMS Health or can be interpreted as Intelligence, “Q” initially comes from Quintiles or can be interpreted as Quotient and VIA is basically the path of transformation]. Since the name of this company can’t be spelt out it remains in capitals in the affiliation section.

2. In ethics statement in the manuscript and in the online submission form, please provide additional information about the patient records/samples used in your retrospective study. Specifically, please ensure that you have discussed whether all data/samples were fully anonymized before you accessed them and/or whether the IRB or ethics committee waived the requirement for informed consent. If patients provided informed written consent to have data/samples from their medical records used in research, please include this information.

Reply: The following text has been added to the manuscript: “Patients are informed of the data collection scheme by the practice and have the ability to opt-out of the database at any time.”

This information is also now explained in the online submission form, along with the reference number of the UK Research Ethics Committee’s approval of the data collection scheme.

Reply: A new supplementary table has been added (S4 Table. Incidence of any types of fatigue and ME/CFS in patients with multiple infections with bacterium Borrelia burgdorferi) to support this statement. Additionally, S3 Table has been added to support an earlier statement (in the first section of the discussion) and provide results related to incidence rates, crude and adjusted hazard ratios of any types of fatigue and ME/CFS for Lyme disease using an extended cohort including also suspected Lyme disease patients and their matched controls.

Reply: We apologize for not doing so in the first version of the manuscript – captions have now been added and citations updated. 

5. Please upload a new copy of Figure 2 as the detail is not clear. Please follow the link for more information: https://blogs.plos.org/plos/2019/06/looking-good-tips-for-creating-your-plos-figures-graphics/" https://blogs.plos.org/plos/2019/06/looking-good-tips-for-creating-your-plos-figures-graphics/

Reply: The Preflight Analysis and Conversion Engine (PACE) digital diagnostic tool has been used to generate a new copy of all figures and these high-resolution figures have now been uploaded.

Reply: The list of references has been carefully reviewed and, although one cited paper triggered a response (reference 15), none seems to have been retracted. A few papers have been added in the reference list – most of them to address the reviewers’ comments. The complete list of newly added papers can be found here:

Carruthers BM, van de Sande MI, De Meirleir KL, Klimas NG, Broderick G, Mitchell T, et al. Myalgic encephalomyelitis: International Consensus Criteria. J Intern Med. 2011 Oct;270(4):327–38. 

Committee on the Diagnostic Criteria for Myalgic Encephalomyelitis/Chronic Fatigue Syndrome, Board on the Health of Select Populations, Institute of Medicine. Beyond Myalgic Encephalomyelitis/Chronic Fatigue Syndrome: Redefining an Illness [Internet]. Washington (DC): National Academies Press (US); 2015 [cited 2021 Nov 2]. (The National Academies Collection: Reports funded by National Institutes of Health). Available from: http://www.ncbi.nlm.nih.gov/books/NBK274235/

Estévez-López F, Mudie K, Wang-Steverding X, Bakken IJ, Ivanovs A, Castro-Marrero J, et al. Systematic Review of the Epidemiological Burden of Myalgic Encephalomyelitis/Chronic Fatigue Syndrome Across Europe: Current Evidence and EUROMENE Research Recommendations for Epidemiology. J Clin Med. 2020 May 21;9(5):E1557. 

Pheby DFH, Araja D, Berkis U, Brenna E, Cullinan J, de Korwin J-D, et al. The Development of a Consistent Europe-Wide Approach to Investigating the Economic Impact of Myalgic Encephalomyelitis (ME/CFS): A Report from the European Network on ME/CFS (EUROMENE). Healthc Basel Switz. 2020 Apr 7;8(2):E88. 

Nemeth J, Bernasconi E, Heininger U, Abbas M, Nadal D, Strahm C, et al. Update of the Swiss guidelines on post-treatment Lyme disease syndrome. Swiss Med Wkly. 2016;146:w14353. 

Gaudino EA, Coyle PK, Krupp LB. Post-Lyme syndrome and chronic fatigue syndrome. Neuropsychiatric similarities and differences. Arch Neurol. 1997 Nov;54(11):1372–6. 

Schutzer SE, Angel TE, Liu T, Schepmoes AA, Clauss TR, Adkins JN, et al. Distinct cerebrospinal fluid proteomes differentiate post-treatment lyme disease from chronic fatigue syndrome. PloS One. 2011 Feb 23;6(2):e17287.

---

## [Editor Report · Decision Letter 1]

8 Mar 2022

Incidence of Lyme disease in the United Kingdom and association with fatigue: a population-based, historical cohort study

PONE-D-21-25917R1

Dear Brellier,

We’re pleased to inform you that your manuscript has been judged scientifically suitable for publication and will be formally accepted for publication once it meets all outstanding technical requirements.

Kind regards,

Utpal Pal, PhD

Academic Editor

PLOS ONE

---

## [Editor Report · Acceptance letter]

14 Mar 2022

PONE-D-21-25917R1 

Incidence of Lyme disease in the United Kingdom and association with fatigue: a population-based, historical cohort study 

Dear Dr. Brellier:

I'm pleased to inform you that your manuscript has been deemed suitable for publication in PLOS ONE. Congratulations! Your manuscript is now with our production department. 

Kind regards, 

on behalf of

Dr. Utpal Pal 

Academic Editor

PLOS ONE